# Site-Coded Oral Squamous Cell Carcinoma Evaluation by Optical Coherence Tomography (OCT): A Descriptive Pilot Study

**DOI:** 10.3390/cancers14235916

**Published:** 2022-11-30

**Authors:** Vera Panzarella, Fortunato Buttacavoli, Alessio Gambino, Giorgia Capocasale, Olga Di Fede, Rodolfo Mauceri, Vito Rodolico, Giuseppina Campisi

**Affiliations:** 1Department of Surgical, Oncological and Oral Sciences, University of Palermo, 90127 Palermo, Italy; 2Department of Surgical Sciences, Oral Medicine Section, CIR Dental School, University of Turin, 10123 Turin, Italy; 3Department of Surgical Sciences, Paediatrics and Gynaecology, University of Verona, Policlinico “G. B. Rossi” of Verona, 37134 Verona, Italy; 4Department ProMISE, University of Palermo, 90127 Palermo, Italy

**Keywords:** optical coherence tomography, early diagnosis, oral cancer, diagnostic pattern, optical biopsy

## Abstract

**Simple Summary:**

Optical Coherence Tomography (OCT) is a non-invasive optical device for diagnostics of epithelial structures, including oral mucosa. To date, there are very few investigations conducted by a methodical comparison between clinical/histological and OCT parameters and with a specific reference to the anatomical site-variability of the oral mucosa. Our study performed an in vivo OCT systematic evaluation of thirty site-coded oral squamous cell carcinomas (in comparison both to the OCT scans of the same site-coded healthy mucosa and to the histological images), to identify potential standardized site-specific OCT-OSCC patterns. This study, representing the first systematic descriptive site-specific OCT investigation of a cohort of OSCCs, aims to support the reliable diagnostic use of OCT in oral cancer.

**Abstract:**

Optical Coherence Tomography (OCT) is an emerging non-invasive method for oral diagnostics, proving to be a practicable device for epithelial and subepithelial evaluation. The potential validity of OCT in oral cancer assessment has been explored but, to date, there are very few investigations conducted with a systematic comparison between clinical/histological and OCT parameters, especially in strict reference to the anatomical site-codification of the oral mucosa. In this regard, our study performed a two-steps evaluation (in vivo OCT and histological investigations) of suspected OSCCs, progressively recruited, using as references the OCT images of the same site-coded healthy mucosa, to provide as much as possible site-specific determinants. Thirty histologically confirmed OSCCs were recruited. Specific OCT mucosal features (SEL—Stratified Epithelial Layer; BM—Basement Membrane; LP—Lamina Propria) were registered and processed using the SRQR (Standards for Reporting Qualitative Research) statement. The systematic dual descriptive OCT analysis revealed that OSCC scans present a complete alteration of epithelial (KL, SEL) and subepithelial (BM, LP) layers with a site-specificity characteristic; moreover, peculiar OCT configurations such as “icicle-like” structures could be strongly suggestive of neoplastic infiltration. This study supports the OCT use for the development of more specific optical structural models applied to oral carcinogenesis.

## 1. Introduction

Oral cancers are the 16th most common cancers overall, the 11th most common cancers in men and the 18th most common cancers in women, accounting for at least >377,700 new cases with a mortality of approximately >17,770 cases worldwide in 2020 [1].

The most common type of oral cancer is the squamous cell carcinoma (SCC) (90% of all oral malignancies) developing, in almost all cases, from oral potentially malignant disorders (OPMDs), such as leukoplakia or lichen planus [2]. Despite the progress in therapy, the mortality of patients with OSCC has remained steadily high during the last 20 years as compared to other cancers and, due to the raising of the demographic age, its global incidence is predicted to increase by almost two-thirds in 2035 [3].

Its early detection and treatment are still crucial to improve the prognosis, and, in this regard, general practitioners and dental professionals are recognized to play an important role in the detection and monitoring of OPMDs and, generally, in the secondary prevention of OSCC.

However, nearly all patients diagnosed with oral cancer at a late-stage report poor routine oral mucosal examination during dental and medical recalls [4]. Furthermore, the extreme clinical heterogeneity of lesions with malignant potential (e.g., ulcer, nodule, erythro-leukoplakia) and the uncontrolled various risk factors (e.g., smoking, alcohol, chronic mechanical trauma, infections, genetics) are responsible for a diagnostic delay estimated at around 12 months for oral SCC, as well as frequent mismanagement [2,4]. There is therefore an extreme need to define and validate tools and techniques, standardized and easy to use, which help in early diagnosis.

The term “optical biopsy” refers to a group of non-invasive diagnostic methods able to give real-time clinical support management for several head and neck pathologies [5]. In oral medicine, the term “optical biopsy” includes a wide range of medical screening tools [5]: autofluorescence, chemiluminescence, elastic scattering spectroscopy (ESS), path length differential spectroscopy (DPS), infrared spectroscopy, Raman spectroscopy, confocal imaging, micro-endoscopy, and coherence tomography.

Among these, Optical Coherence Tomography (OCT) is a diagnostic imaging method, first applied in 1991 by Huang et al. [6] in ophthalmology: it is a non-invasive imaging technique analogous to ultrasound, that measures the amplitude of backscattered light generated from a light source as a function of depth. OCT permits in vivo, non-invasive imaging of microscopic characteristics of skin and mucosal structures, and it was proved to be a valid and practicable method for the determination of epithelial structure.

The potential validity of OCT in oral medicine was widely investigated [7,8,9,10,11,12,13] and several ex vivo and in vivo studies compared the OCT images of normal mucosa with those with suspected oral squamous cell carcinoma (OSCC) lesions providing useful OCT potential diagnostic indicators of progressive tissue transformation from normal epithelium to early invasive carcinoma in the oral cavity [14,15,16,17].

However, to date, there does not exist a “bank of OCT in vivo images” of OSCC to consent the immediate assessment of OSCC by this device (as for skin cancers) [18,19], especially because the majority of the investigations on this topic are case reports/series ex vivo studies, or else conducted through the use of OCT prototypes and/or of not yet standardized and easily reproducible diagnostic algorithms, such as that proposed by some authors [20]. Moreover, the OCT evaluation of oral lesions is too often performed without a site-definition of oral mucosa and/or without a systematic comparison both with the corresponding histological images and the OCT images of the same healthy mucosa site. Especially this last aspect would seem to play a very crucial role in the interpretation difficulty of the lesions by OCT, since the epithelial macrostructure, presents a high physiological difference among the distinct oral sites (i.e., mobile tongue, gum, palate, floor of the mouth) [3].

Against these critical background issues, the scientific community should make a greater effort to define studies adequately supported by the comparative evaluation between oral site-coded healthy and lesions of the mucosa to provide as many site-specific OCT determinants as possible for oral malignancy.

In this regard, our non-invasive study aimed to perform a focused interpretation of in vivo OCT OSCC set images, using as references the OCT images of the same site-coded healthy mucosa and through comparison with the site-specific histological confirmatory examination of the same lesion.

## 2. Materials and Methods

The study protocol conformed to the ethical guidelines of the 1964 Declaration of Helsinki and its later amendments or comparable ethical standards. It was also approved by the Institutional Review Board of University Hospital Policlinico “Paolo Giaccone” in Palermo (Italy) (approval number 11/2016).

### 2.1. Study Design

This is a qualitative research study and adheres to the to the EQUATOR guidelines of reporting research using the Standards for Reporting Qualitative Research (SRQR) checklist for instrumental and clinical data collection and processing [21]. Given the exploratory nature of the study, no preliminary sample measurement or comparative evaluation of the statistical difference between the categorical variables were carried out.

### 2.2. Entry Criteria

The recruitment of subjects started on 1 October 2021 and finished on 30 September 2022. All participants, after written informed consent, were consecutively recruited, at the Oral Medicine Unit of the University Hospital Policlinico “Paolo Giaccone” in Palermo (Italy). Data on gender and age were recorded.

The eligibility criteria were:Age ≥ 18 yearsAbility to provide informed consentClinical diagnosis strongly suggestive of OSCCNo previous history of OSCC or previous anti-cancer therapy for OSCC

### 2.3. OSCCs Evaluations

Clinical evaluation. Patients with entry criteria were progressively recruited and screened by the same oral medicine expert. One digital photograph per lesion was made to record the site in which both the OCT and histological examinations were performed. All photographs were taken by using Nikon D7200 Camera, with Nikon AF-S DX 105 mm F2.8G Lens and Nikon R1C1 dual flash.

OCT evaluation. For this study, we used the device OCT SS-OCT VivoSight^®^, Michelson Diagnostics Ltd., version 2.0, Orpington, Kent, UK. The system type is a Swept-source Fourier-Domain OCT. The light source of the device is a Santec HSL-2000-12 wide sweep laser with. The laser center wavelength is 1305 ± 15 nm and the laser frequency sweep range is ≥150 nm. The axial optical resolution in tissues is <10 μm and the lateral resolution is <7.5 μm, with a maximum scan width of 6 mm × 6 mm to a focal depth of ≈2 mm. For the OCT evaluation the same protocol (and limitations) detailed in our previous study [22] was applied. OCT evaluation was performed for each lesion and the most representative OCT images were archived.

Each lesion was site-coded applying the 2021 NIH/SEER ICD-0-3.2 topographical classification codes (from C02.0 to C02.2 for the tongue, C03.0 and C03.1 for the upper and lower gum, respectively and C06.0, for cheek mucosa, buccal mucosa, and internal cheek) [23,24]. For each selected OCT image of suspected OSCC, an OCT image of the same site-coded healthy mucosa from volunteers was recovered. Healthy subjects were randomly selected independent of their age and sex and were recruited in our department for the treatment of other, non-malignant mucosal diseases.

OCT analysis was performed using a set of criteria concerning Keratinized Layer (KL), Stratified Epithelial Layer (SEL), Basement Membrane (BM) and Lamina Propria (LP), as shown in Table 1. OCT images were evaluated by two different OCT examiners (V.P. and F.B.) and if the two examiners disagreed, a third examiner (Gio.C.) assessed the OCT images.

Histological evaluation. To confirm clinical OSCC suspicion, histological examination was carried out on all patients. After local anesthesia, an incision was performed with a 6 mm diameter punch biopsy in the same OCT scanned site, appropriately marked with a skin pencil (Dima, cd. 33176).

The biopsy specimens were processed routinely, fixed in 10% neutral buffered formalin solution, and embedded in paraffin, and sent to Pathology to perform histopathological OSCC diagnosis. Formalin-fixed, paraffin-embedded (FFPE) sections of 5 µm were stained with routine hematoxylin and eosin and examined to confirm the diagnosis. The pathologist (VR) examined the images independently and in blindness from clinical and OCT diagnoses. Finally, the OCT and histological image data for each lesion were compared to discriminate potentially concordant diagnostic patterns.

## 3. Results

Thirty patients with oral squamous cell carcinoma, histologically diagnosed, were selected, and categorized by NIH/SEER ICD-0-3.2 topographical classification codes, as detailed below.

-six case of OSCC on the dorsal surface of the tongue (code C02.0)-six cases of OSCC on the border of the tongue (code C02.1)-five case of OSCC on the ventral surface of the tongue (code C02.2)-six cases of OSCC on anterior alveolar mucosa (code C03.1)-seven cases of OSCC on the buccal mucosa (code C06.0)

The main characteristics of OSCC sites, gender, and age (reported in mean value) are shown on Table 2.

For each of the five different OSCC sites investigated, a corresponding OCT image of healthy mucosa was selected for comparative analysis.

Figures 1–5 show clinical, OCT and histological images acquired from the most representative case collected by site. For the OCT evaluation, OSCC OCT images were compared first with the OCT reference images achieved from the same site-coded healthy oral mucosa, then with histological images. From these progressive comparisons, the following recurring structural similarities emerged.

### 3.1. Clinical, OCT and Histopathology Images of OSCC on the Dorsal Surface of Tongue Compared to the Same-Site Healthy Mucosa (Site Code C02.0)

The OCT image of the healthy dorsal surface of the tongue (Figure 1a,b) denotes the superficial physiological irregularity of specialized tissue. The papillae (F) assume a characteristic hyper-reflective oval-round image, in which it is not possible to discriminate both the overlying KL and the underlying SEL. Due to the limited penetration of the light, the subpapillary area results in hypo-reflective appearances (#), if compared to the contiguous interpapillary sections. Moving to the sub-epithelial layer, it is possible to evaluate a regular reflectivity both of BM (continuously assessable) and of LP that appears as a deeper hyper-reflective well demarcated and distinguishable band from SEL.

For dorsal tongue OSCC (Figure 1c), OCT image highlighted the evident loss of the typical aspect of healthy specialized mucosa, without differences, among tissue layers, in light reflection. The appearance of the normal lingual papillae is completely indistinguishable, and BM is not assessable, while LP is indistinguishable for SEL (Figure 1d).

From LP, hyper-reflective cords deepen in the surrounding lower tissue (Figure 1d). These structures are like those described in the literature for OCT images performed on skin melanomas that have been called “icicle-like” structures and could denote neoplastic tissue infiltration [25,26].

The diffuse tissue disorganization, as well as the superficial ulcerations, is clearly visible in OCT and histological images (Figure 1d,e).

### 3.2. Clinical, OCT and Histopathology of OSCC on Lateral Borders of Tongue Compared to the Same-Site Healthy Mucosa (Site Code C02.1)

In OCT images of lateral borders of the healthy tongue (Figure 2a,b) the keratinized layer (KL) is continuously assessable as a well demarcated hyper-reflective band from hypo-reflective underlying SEL. The BM appears intact at every point of the tomographic section. The LP has a peculiar streaking aspect, reflecting the connective organization (collagen bundles).

On border tongue OSCCs (Figure 2c,f), the sharp demarcation of all epithelial and subepithelial layers is not assessable. The local invasiveness of the tumor could be evaluated by the presence of “icicle-like” structures, also seen in these OCT images (Figure 2d,g). Additionally, for these OSCCs, the different morphological aspects, ulcerative (Figure 2c) vs. exophytic (Figure 2f), are evident both in OCT and in histological images (Figure 2d,e,g,h, respectively).

### 3.3. Clinical, OCT and Histopathology Images of OSCC on Ventral Surface of Tongue Compared to the Same-Site Healthy Mucosa (Site Code C02.2)

In the OCT image of the healthy ventral mucosa of the tongue (Figure 3a,b) the KL is physiologically absent. The SEL appears as a homogeneous and hypo-reflective band; both BM and LP are well demarcated from SEL. Within LP, hyper-reflective and lax reticular structure reflecting the organization of the connective fibers, surrounding hyper-reflective blood vessels (V).

For the ventral tongue OSCC (Figure 3c), OCT highlighted the evident loss of tissue architectural pattern with no demarcation between SEL, BM and LP. In this OSCC case, superficial epithelial ulcerations are present in OCT scan (Figure 3d) and simultaneously evaluable in the histopathologic image, due to the neoplastic tissue spread to the surface associated with superficial fibrin deposits (Figure 3e).

### 3.4. Clinical, OCT and Histopathology of OSCC on Low Anterior Alveolar Mucosa and Mandibular Gingiva Compared to the Same-Site Healthy Mucosa (Site Code C03.1)

In OCT images of low anterior healthy alveolar mucosa and mandibular gingiva (Figure 4a,b), the superficial KL appears as a thinner and more reflective band on the alveolar mucosa, compared to gingival mucosa (Figure 4b) and SEL is constantly assessable and hypo-reflective. BM appears continuously intact, and LP is perfectly demarcated with characteristically mottled in a linear pattern.

On the OSCC images from these sites (Figure 4c,f,i), the OCT pattern appears deeply distorted: it is impossible to distinguish between SEL and LP, and BM is absent (Figure 4d,g,l). The OCT examination also showed the presence of superficial hypo-reflective areas (Figure 4d); such zones of epithelial flaking could likely be expressions of a microbial infection detectable also on the histological preparation in the superficial areas markedly colored in blue (Figure 4e) because of the higher affinity of the microorganisms with the hematoxylin dye.

In the specific case of OSCC on IV sextant (Figure 4f), it is very significant to observe the OCT tissue transition (Figure 4g), from the healthy mucosa (right side of the image), where the SEL, BM and the sub-epithelial tissue are distinguishable, to OSCC portion (left side of the image), where the progressive disappearance of BM and consequent impossibility of a clear distinction between SEL and LP, are evident.

These transformative characteristics can be peculiar and denote malignant connotations even in the absence of “icicle-like” structures. The lack of the latter could be precisely attributed to the absence of a marked neoplastic tissue infiltration in the selected OCT section/image, compared to the histological section chosen for the diagnostic confirmation of OSCC (Figure 4h).

In the exophytic OSCC on V sextant (Figure 4i), histological images show a pushing superficial exophytic growth, representative of verrucous OSCC variant (Figure 4m); the same layout is clearly assessable in the corresponding OCT scan (Figure 4l).

Additionally, in the two cases on mandibular gingiva, OCT assesses the “icicle-like” structures (Figure 4d,l).

### 3.5. Clinical, OCT and Histopathology Images of OSCC on Buccal Mucosa Compared to the Same-Site Healthy Mucosa (Site Code C06.0) Aspect

In the OCT images of the healthy buccal mucosa (Figure 5a,b), SEL appears as a hypo-reflective homogeneous band under a dim KL layer. LP shows as a thinly mottled hyper-reflective network and BM maintains the whole architectural integrity.

OCT and histological images of buccal mucosa OSCCs (Figure 5c–s), display loss of the normal epithelial and subepithelial stratifications; with the constant presence of hyper-reflective “icicle-like” structures that deepen from the superficial layers in the surrounding hypo-reflective tissue.

Interesting is the relief, in the OCT images (Figure 5d,g,l,o), of the details relating to the fibrin deposits, detectable as mottled hypointense areas upper the superficial ulcerations.

## 4. Discussion

In the last two decades, OCT has shown great potential in the preliminary screening, diagnosis, and monitoring of OPMDs and OSCC. However, very few studies investigated oral carcinogenesis in relation to physio-pathological changes of oral mucosa, from healthy to cancer [27,28] and, to the best of our knowledge, no OCT analysis was performed respecting an oral cavity site-specificity. Moreover, recent systematic reviews suggest an OCT data interpretation that is strongly user-dependent, without standardization of OCT oral malignancy indicators, correctly compared with histological confirmatory investigation [29].

The present pilot descriptive study has considered all these critical issues in a cohort of 30 patients with site-coded OSCC, histologically diagnosed. The used in vivo OCT system setup (the same applied for skin imaging evaluation) [30] provided a focal depth equal to 2 mm and a lateral/axial resolution <10 μm. These properties allowed it to clearly discriminate between healthy and pathological tissue, identifying specific characteristics of the epithelial (KL, SEL) and subepithelial (BM, LP) layers.

By OCT evaluation of oral healthy mucosa, KL appears clearly visible, as a strongly hyper-reflexive continuous superficial band, in the keratinized tissue; this aspect is particularly evident for healthy alveolar mucosa and for gingiva. On the contrary, in site physiologically variably or not keratinized (i.e., lateral borders of the tongue, buccal mucosa), KL is normally less evaluable. On the lingual dorsum, KL is not distinguishable from the underlying layer because of the presence of lingual papillae and consequent irregularity diffusion of OCT light.

SEL appears as a constant homogeneous hypo-reflection band area, well assessable with delimiting layers; apart from the lingual dorsum, where the characteristic intermittent hyper-reflective oval-round aspect of the papillae do not allow for discrimination between both the overlying KL and the underlying SEL.

BM is constantly present in all healthy tissue, appearing as a clear dividing line between the SEL and underlying LP. Although, at the lingual dorsum, the BM is alternately hypo-reflective, its integrity is continuously scanned by OCT at every point.

As regards the LP, despite its distinction from SEL at any sites, it presents peculiar regional differences, able to make it an OCT parameter with the greater site-variability and inhomogeneity in optical density. Particularly at the dorsal tongue, LP appears discontinuously darker, due to the presence of papillae limiting the effective imaging scanning depth and, at the ventral surface of the tongue, LP assumes a lax reticular structure surrounding major and minor blood vessels. At the other sites, LP appears as a typical linear pattern reflecting the physiologically tissue organization.

All the details described for the parameters listed above are also reported in similar studies investigating the OCT discriminatory potential of healthy oral mucosa alone [3,14,31,32,33,34].

About OCT-OSCC evaluation, in our study, the normal patterns previously analyzed result completely upset, with both common and strictly distinct, site-specific characteristics. All different epithelial and subepithelial layers no longer maintain their integrity and their optical light characteristics, resulting completely altered from the OCT analysis.

KL, SEL and LP are not assessable in OSCC scans; optical density and light diffusion assume a pathological homogeneity in all investigated sites, especially in comparison with SEL and LP. These characteristics are even more evident in those sites normally characterized by a peculiar physiological morphology, in terms of specialized mucosa (i.e., lingual papillae in dorsal surface of the tongue) and/or of intimate continuity with specific subepithelial structures (i.e., blood vessels in the ventral surface of the tongue): these peculiarities are completely indistinguishable on OCT analysis, as well as in histological microscopic evaluations.

In this study, the ability of OCT to intercept architectural changes indicative of cancer is evident in the analysis of the contiguous tissues from healthy to pathological sites (e.g., Figure 4g) where the BM integrity is gradually lost; its absence is strongly discriminatory of invasive neoplastic subepithelial invasion, as found by comparison with the histological images; moreover, previous studies, applying OCT in oral oncology, have reported the loss of BM as a discriminatory criterion between healthy and cancerous tissues.

Ridgway et al. [35], for the first time, observe that the discontinuity and violation of BM can be observed by OCT imaging related to malignant transformation in association with other different epithelial tissue microanatomy changes, resulting in an irregular stratification, such as epithelial thickening, proliferation, invasion and broadening of rete pegs. Other authors reported the importance of several parameters to distinguish between benign and malignant oral lesions, such as disorganization of epithelial stratification, epithelial thickening, and the integrity of the BM [14,15,17,27]. More recently, the BM alteration has also been applied to automated OCT image processing methods with promising results. Particularly, Heidari et al. [36] compared the diagnostic efficacy of clinical visual scoring and an automated diagnostic algorithm, analyzing the depth-resolved intensity distribution and vertical deviation in the BM; the image-processing algorithm showed a high diagnostic accuracy with agreement to the histopathological gold standard in distinguishing healthy cases from cancer and dysplasia (sensitivity, 95%; specificity, 100%) and cancer from dysplasia (sensitivity, 91%; specificity, 100%).

In the present study, OCT has demonstrated a specific discriminative potential to identify ulcerative and exophytic morphology of OSCCs, visible as areas of cancer cells superficializing with superlayer fibrin debris and highly hyper-reflexive surface offshoots, respectively. Particularly for the exophytic OSCC form, OCT seems also to be able to recognize verrucous microscopic variants, identified as characteristic superficial exophytic growth zones. All these macro- and micro-OSCC features are confirmed by histopathology.

Another OCT peculiarity, emerged from our study, is the common report, in almost all scanned OSCC sites, of hyper-reflective conical structures that deepen from the superficial cellular layers (SEL) to the deeper ones (BM and LP). This texture was named “icicle-like” by the authors, because of its similarity with architectural frames shown for melanoma and malignant skin lesions [25,26], called “icicle-shaped” structures.

Gambichler et al. suggest that the icicle structure is an expression of the vertical infiltration phase by dermal layers of neoplastic melanocytes. Garbarino et al., in a retrospective analysis aimed to identify correlating features observed with reflectance confocal microscopy and OCT for different nodular skin lesions, reported this “icicle-shaped” structure in almost all cases (17/18) of melanomas investigated with the similar OCT device to ours (VivoSight” D-OCT, Michelson Diagnostics Orpington, Kent, UK.). According to the authors, the interpretation of these structures is that of a “dense infiltrate of lymphocytes and cancerous cells, exclusively attributed to melanoma” [26] However, in the same study, the authors hypothesize that this feature could also be justified by the presence of darker hypo-reflective “shadow cones”, intermixed with areas of greater light reflection (the icicle structure) and produced by cerebriform nests in the underlying portion of the OCT image [26].

The strong similarity between the “icicle-like” structures, detected by OCT in our OSCCs and those of other authors in melanomas, could suggest the mutual concept of front of tumor squamous infiltration towards the oral subepithelial layers. At the same time, also for OSCC, this OCT layout seems to be strongly related to lesion microstructure: more expressed in exophytic/nodular forms (where light diffusion is intermittent) rather than ulcerative/erosive morphology (where light stratification is more homogeneous).

The results of the present study confirm the ability of OCT to discriminate the epithelial and subepithelial layers of oral mucosa in different sites and to recognize site-specific OSCC alteration, histologically supported. A comprehensive proposal of site-specific standardized OCT patterns for healthy tissues compared to the OSCC are summarized in Table 3.

Some limitations of the present study could be recognizable. The first is the use of an OCT probe for dermatological assessment, which was not able to evaluate all sites in the mouth, such as soft and hard palate, retromolar region, and floor of mouth. However, authors discriminated all the other mucosal sites with similar microscopic features to the missing sites in terms of epithelial differentiation features (keratinized vs. non keratinized).

Another potential limit could be the absence of the data on measuring the keratin layer and thickness of epithelium, as indicated by the previous studies as important in OCT carcinogenesis evaluation. Indeed, this represents a specific choice rather than a limitation of our study. Hamdoon et al. [15] reported that the thickness of the mucosa layers alone is not conclusive for the OCT diagnosis formulation and other features of the epithelial and subepithelial are necessary. In addition, we believe that proposing a method of standardization of OCT-OSCCs expressed in specific descriptive characteristics (as standardized as possible), rather than in measurements of tissue layers (difficult to reproduce), can allow using the device more easily, even for specialists outside of oral medicine.

It is well known that the interpretation of OCT images is strongly related to operator-ability in performing the examination and reading images. To overcome this disadvantage, “trained” machine learning systems have recently been proposed to ensure more diagnostic certainty by OCT, such as successfully validated in the ophthalmology and skin fields [37]. For oral tissue characterization, very few studies investigated computational algorithms to perform potential automated oral cancer diagnosis from OCT images, and only two investigations using in vivo methods [20,36] based prevalently on BM evaluation and epithelial thickness score, respectively. Their results are encouraging, and we hope, soon, to provide, thanks to a huge number of cases confirmed, similar “trained” learning models, considering all the epithelial and subepithelial layers, systematically compared and analyzed like in this study, and validated on a larger and multicentric sample.

## 5. Conclusions

The results of this study confirm that epithelium (KL and SEL), basal membrane (BM) and lamina propria (LP) can be clearly identified with OCT in oral mucosa with a specific detail by site. Moreover, loss of normal characteristics of the oral mucosa layers seen in OSCCs, compared to physiological images and histological investigation, can be standardized and site-coded, supporting the diagnostic process and the development of more specific optical structural models applied to oral carcinogenesis.

## Figures and Tables

**Figure 1 cancers-14-05916-f001:**
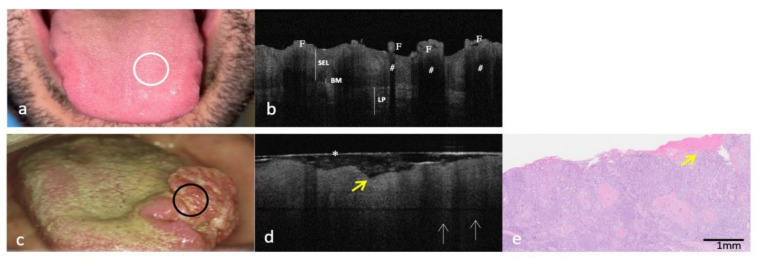
Clinical, OCT and histopathology images of a case of OSCC on the dorsal surface of the tongue compared with the same healthy site (code C02.0). (**a**,**b**): Clinical and OCT images of healthy mucosa of dorsal surface of tongue; the site of in vivo OCT evaluation is indicated with a white circle. (**c**–**e**): Clinical, OCT and histopathological images (H-E stain; original magnification ×25) of OSCC on dorsal surface of tongue; the sites of in vivo OCT evaluation are indicated with a black circle; “icicle-like” structures are indicated with white arrows; superficial ulcerations are indicated by yellow arrows. An asterisk (*) indicates the thin transparent plastic wrapping around the scanning probe F—Lingual papillae; #—Subpapillary spaces; SEL—Stratified Epithelial Layer; BM—Basement Membrane; LP—Lamina Propria.

**Figure 2 cancers-14-05916-f002:**
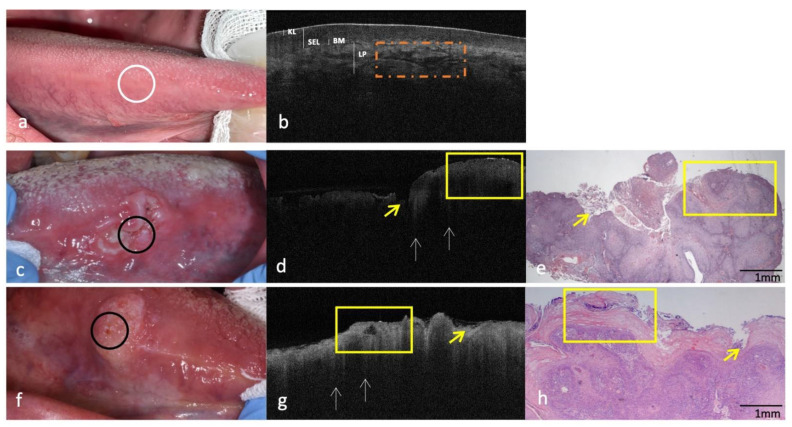
Clinical, OCT and histopathology images of two cases of OSCC on the lateral borders of the tongue compared with the same healthy site (code C02.1). (**a**,**b**): Clinical and OCT images of healthy mucosa of the border of tongue; the site of in vivo OCT evaluation is indicated with a white circle. (**c**–**h**): Clinical, OCT and histopathological images (H-E stain; original magnification ×25) of two OSCCs on the border of tongue; the site of in vivo OCT evaluation is indicated with a black circle; “icicle-like” structures are indicated with white arrows; superficial ulcerations and exophytic morphology are indicated by yellow arrows and yellow box, respectively; streaking pattern of LP is indicated by a dotted orange box. KL–Keratinized layer; SEL—Stratified Epithelial Layer; BM—Basement Membrane; LP—Lamina Propria.

**Figure 3 cancers-14-05916-f003:**
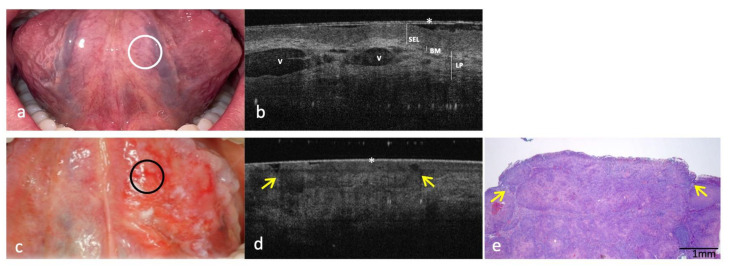
Clinical, OCT and histopathology images of a case of OSCC on the ventral surface of the tongue compared with the same healthy site (code C02.2). (**a**,**b**): Clinical and OCT images of healthy mucosa of the ventral surface of the tongue; the site of in vivo OCT evaluation is indicated with a white circle. (**c**–**e**): Clinical, OCT and histopathological images of (H-E stain; original magnification ×25) confirmed OSCC on the ventral surface of tongue; the site of in vivo OCT evaluation is indicated with a black circle. An asterisk (*) indicates the thin transparent plastic wrapping around the scanning probe; superficial ulcerations are indicated by a yellow arrow. SEL—Stratified Epithelial Layer; BM—Basement Membrane; LP—Lamina Propria; V–vascular structures of sublingual area.

**Figure 4 cancers-14-05916-f004:**
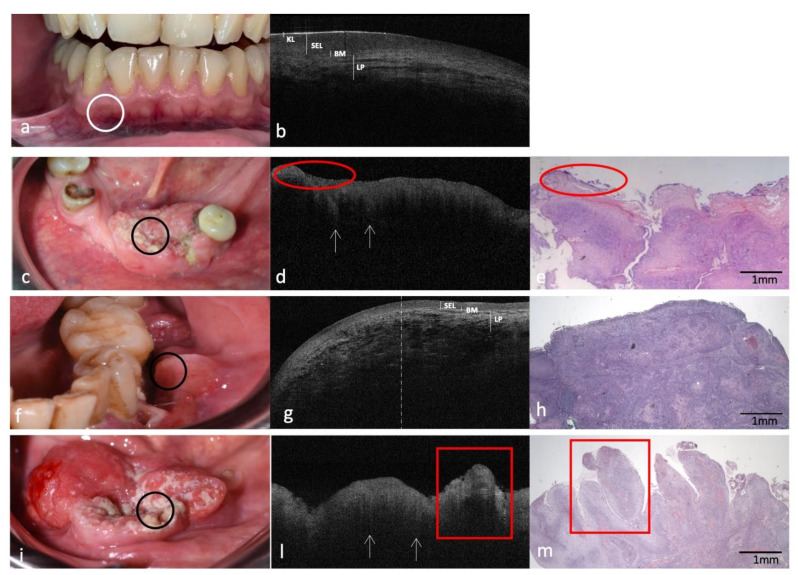
Clinical, OCT and histopathology images of three cases of OSCCs on low anterior alveolar mucosa and mandibular gingiva compared with the same healthy site (code C03.1). (**a**,**b**): Clinical and OCT images of healthy mucosa of lower alveolar mucosa and gingiva; the site of in vivo OCT evaluation is indicated with a white circle. (**c**–**i**,**l**,**m**): Clinical, OCT and histopathological images (H-E stain; original magnification ×25) of three OSCCs on lower anterior and left alveolar mucosa; the site of in vivo OCT evaluation is indicated with a black circle; “icicle-like” structures are indicated with white arrows. Microbial proliferation on epithelial flaking is indicated by red ovals; the exophytic growth zones of the verrucous variant of OSCC are indicated by red boxes; a dashed line in (**g**) separates the right (healthy) from the left (OSCC) portions of the image. KL–Keratinized layer; SEL—Stratified Epithelial Layer; BM—Basement Membrane; LP—Lamina Propria.

**Figure 5 cancers-14-05916-f005:**
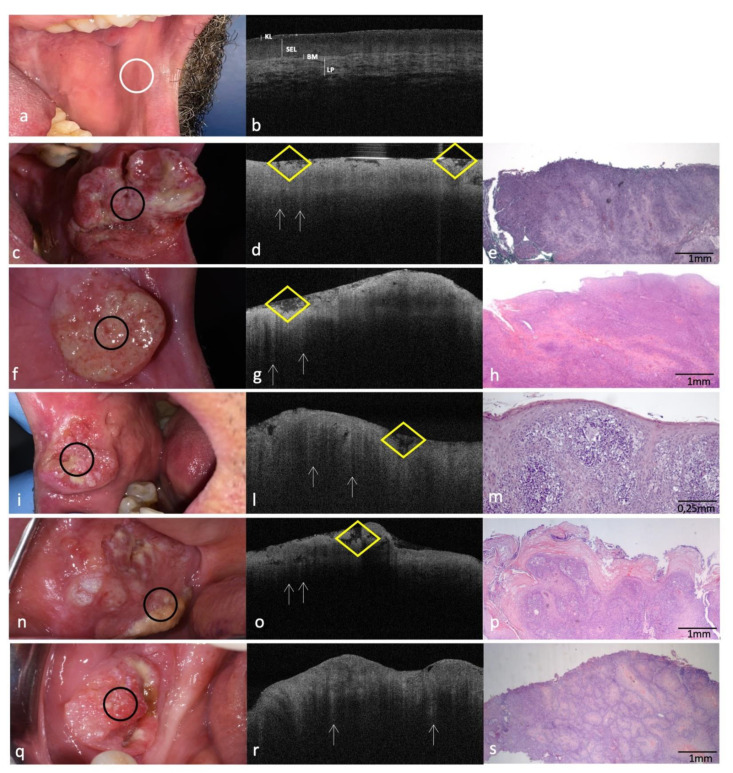
Clinical, OCT and histopathology images of OSCCs sited on buccal mucosa compared with the same healthy site (code C06.0). (**a**,**b**): Clinical and OCT images of healthy buccal mucosa; the site of in vivo OCT evaluation is indicated with a white circle (**c**–**i**,**l**–**s**): Clinical, OCT and histopathological images (H-E stain; original magnification ×25 in (**e**,**h**,**p**,**s**); original magnification ×100 in (**m**)) of five OSCCs on buccal mucosa; the site of in vivo OCT evaluation is indicated with a black circle; “icicle-like” structures are indicated with white arrows; fibrin deposits are indicated with a yellow rhombus. KL–Keratinized layer; SEL—Stratified Epithelial Layer; BM—Basement Membrane; LP—Lamina Propria.

**Table 1 cancers-14-05916-t001:** OCT parameters for oral epithelial and subepithelial evaluations.

Parameters of the OCT Image Analysis	
Parameter	Evaluation
**Keratin layer (KL)**	Assessable/Hyper-reflective Not assessable
**Squamous epithelial layer (SEL)**	Assessable/Hypo-reflective Not assessable
**Basement membrane (BM)**	Continuously assessable Discontinuously assessable Not assessable
**Lamina propria (LP)**	Well demarcated/Distinguishable from SEL Not demarcated/Indistinguishable from SEL

**Table 2 cancers-14-05916-t002:** Demographics data (gender and age) for OSCCs groups by sites.

OSCC Cases by Code	Female (*n*)	Male (*n*)	Total (*n*)	Mean Age (years)
**C02.0 (tongue dorsum)**	3	4	6	75.3
**C02.1 (tongue lateral border)**	4	3	6	65.3
**C02.2 (tongue ventral surface)**	3	2	5	67.4
**C03.1 (anterior inferior alveolar mucosa)**	2	4	6	74.8
**C06.0 (buccal mucosa)**	3	4	7	65.6
**Total**	15	15	30	69.6

**Table 3 cancers-14-05916-t003:** Site-specific patterns of OCT images for healthy tissues compared to the same site localized OSCC.

Parameter	Healthy Tissue OCT	OSCC Lesion OCT
	**Dorsal surface of tongue** (site code C02.0)	
**KL**	Physiological not assessable	Not assessable
**SEL**	Assessable and physiologically irregular Papillae visualized as hyper-reflective oval round reliefs	Not assessable and pathologically homogeneous to light reflection Papillae are completely indistinguishable
**BM**	Continuously assessable	Not assessable
**LP**	Well demarcated/Distinguishable from SEL	Not demarcated/Indistinguishable from SEL Presence of “icicle-like” structures
	**Lateral border of tongue** (site code C02.1)	
**KL**	Assessable	Not assessable
**SEL**		Not assessable
	Assessable/Hypo-reflective	

**BM**	Continuously assessable	Not assessable
**LP**	Well demarcated/Distinguishable from SEL Streaking pattern	Not demarcated/Indistinguishable from SEL Presence of “icicle-like” structures
	**Ventral surface of tongue** (site code C02.2)	
**KL**	Physiologically not present	Not present
**SEL**	Assessable/Hypo-reflective	Not assessable
**BM**	Continuously assessable	Not assessable
**LP**	Well demarcated/Distinguishable from SEL Hyper-reflective and lax reticular structure surrounding blood vessels	Not demarcated/Indistinguishable from SEL
	**Inferior alveolar mucosa and gingiva** (site code C03.1)	
**KL**	Assessable and continuously hyper-reflective	Not assessable
**SEL**	Assessable/Hypo-reflective	Not assessable
**BM**	Continuously assessable	Not assessable
**LP**	Well demarcated with linear-mottled pattern	Not demarcated/Indistinguishable from SEL Presence of “icicle-like” structures
	**Buccal mucosa** (site code C06.0)	
**KL**	Physiologically less assessable	Not assessable
**SEL**	Clear and continuous hypo-reflective homogeneous band-shaped area	Not assessable
**BM**	Continuously assessable	Not assessable
**LP**	Well demarcated/Distinguishable from SEL Mottled hyper-reflective network	Not demarcated/Indistinguishable from SEL Presence of “icicle-like” structures

KL–Keratinized layer; SEL—Stratified Epithelial Layer; BM—Basement Membrane; LP—Lamina Propria.

## Data Availability

Data available on request.

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
