# Peer review of "Site-Coded Oral Squamous Cell Carcinoma Evaluation by Optical Coherence Tomography (OCT): A Descriptive Pilot Study"

_cancers, 2022, doi:10.3390/cancers14235916_

Round 1

Reviewer 1 Report

The article reports on very interesting avenue of research. The authors are just on the start of their journey.  The tumours shown in the article where obvious scc.  It will be interesting if this technology can be used to scan white patches to look for areas of degeneration and malignant potential.  The images see to lend themselves to AI evaluation in due course.

Author Response

Review report:

The article reports on very interesting avenue of research. The authors are just on the start of their journey.  The tumours shown in the article where obvious scc.  It will be interesting if this technology can be used to scan white patches to look for areas of degeneration and malignant potential.  The images see to lend themselves to AI evaluation in due courses.

Response. Thank you for kind comments. We are absolutely in agreement with your thinking, and we believe that a methodological sequence as standardized as possible (such as the one we tried to define for this OSCC sample) is the only strategy for the effective use of OCT in the field of oral diagnostics and for the implementation of any related-targeted learning algorithms.

A complete minor revision of the English, as requested, was performed.

Reviewer 2 Report

Dear Authors

This manuscript titled “Site-Coded Oral Squamous Cell Carcinoma Evaluation By Optical Coherence Tomography (OCT): A Descriptive Pilot Study” is a well thought out study and adds significant knowledge to the existing literature in oral squamous cell carcinoma by optical coherence tomography (OCT). However, the manuscript requires the following clarifications and changes before publication.

Materials & Methods

1.    Where the OSCC cases and the healthy subjects age and gender matched?

2.    It is suggested to include a few more sentences in the materials and method section about the healthy subjects that you have used in this study.

3.    Were the OCT images evaluated by more than one examiner?

4.    What was the fixative used. Is it 10% formalin solution or 10% neutral buffered formalin solution?

Results

5.    For all photomicrographs kindly add the measurement bar.

6.    Figure 1d & e: The yellow arrow indicating the ulcer does not correlate between the OCT image and the photomicrograph. The yellow arrow might have to be moved to the right in the photomicrograph. Kindly look into this. If you are moving the yellow arrow to the right in the photomicrograph then you might need to provide with a new photomicrograph which is representative of the OCT image. Kindly factor in the shrinkage that might have taken place when the tissues were fixed with the fixative solution.

7.    Figure 2b: “The LP has a peculiar streaking aspect, reflecting the connective organization (collagen bundles)”. Hope this can be indicated with arrows in the OCT image for better understanding to the readers who are not experts in this field.

8.    Figure 4g: “It is very significant to observe the OCT pathological tissue transition (Fig. 4 g), from the healthy mucosa (right side of the image), where the SEL, BM and the sub-epithelial tissue are distinguishable, to OSCC portion (left side of the image)”. Hope the authors can explain further the features that are distinguishable between the healthy mucosa and the OSCC portion.

9.    Figure 4g: The "icicle-like" structures are missing which is indicative of OSCC. Hope the authors can explain this finding.

10. Figure 5e,h,p: “fibrin deposits are indicated with a yellow rhombus”. Unfortunately, I am unable to appreciate the fibrin deposits in these photomicrographs. May be a photomicrograph of higher magnification of these regions can be provided as inserts within these figures.

Discussion:

11. “Already previous studies, applying OCT in oral oncology, have reported the loss of BM as a discriminatory criterion between healthy and cancerous tissues”. Kindly merge this sentence with the paragraph below. Kindly give reference by citing Obade et al 2021 and other articles for this sentence.

12. “In addition, we believe that proposing a method of standardization of OCT-OSCCs expressed in specific descriptive characteristics (as standardized as possible), rather than in measurements of tissue layers (difficult to reproduce), can allow using the device more easily, even from non-oral medicine specialists”

Kindly change it to “for other specialist other than oral medicine”.

Thank you

Author Response

Response to Reviewer 2 Comments

Review report:

This manuscript titled “Site-Coded Oral Squamous Cell Carcinoma Evaluation By Optical Coherence Tomography (OCT): A Descriptive Pilot Study” is a well thought out study and adds significant knowledge to the existing literature in oral squamous cell carcinoma by optical coherence tomography (OCT). However, the manuscript requires the following clarifications and changes before publication.

Materials & Methods

  1. Where the OSCC cases and the healthy subjects age and gender matched?
  2. It is suggested to include a few more sentences in the materials and method section about the healthy subjects that you have used in this study.

Responses 1-2. Thank you for your suggestions on implementing information related to healthy OCT mucosa images. Appropriate details on the recruitment of subjects have been entered both in M&M (as you suggest, lines 143-145) and in Results (lines 175-176), to provide the exact correspondence of the method used.

  1. Were the OCT images evaluated by more than one examiner? 
  2. What was the fixative used. Is it 10% formalin solution or 10% neutral buffered formalin solution?

Responses 3-4.  Thank you for pointing out this lack of details. Unfortunately, they represent some writing forgetfulness, and now we have proceeded to insert correctly in the appropriate parts (lines 148-149 and 157, respectively).

Results

  1. For all photomicrographs kindly add the measurement bar. 

Response 5. Thank you for the request. Measurement bar has been added to photomicrographs as required.

  1. Figure 1d &e: The yellow arrow indicating the ulcer does not correlate between the OCT image and the photomicrograph. The yellow arrow might have to be moved to the right in the photomicrograph. Kindly look into this. If you are moving the yellow arrow to the right in the photomicrograph then you might need to provide with a new photomicrograph which is representative of the OCT image. Kindly factor in the shrinkage that might have taken place when the tissues were fixed with the fixative solution.

Response 6. Thank you for the request for greater correspondence between OCT and histological images. Unfortunately, the position of the indicator arrow was incorrect, and has been appropriately modified as required.

  1. Figure 2b: “The LP has a peculiar streaking aspect, reflecting the connective organization (collagen bundles)”. Hope this can be indicated with arrows in the OCT image for better understanding to the readers who are not experts in this field.

Response 7. Thanks for the request. Following your advice, we have specified/indicated this characteristic aspect of the LP both in Figure 2b (with a dotted orange box) and in the relative caption

  1. Figure 4g: “It is very significant to observe the OCT pathological tissue transition (Fig. 4 g), from the healthy mucosa (right side of the image), where the SEL, BM and the sub-epithelial tissue are distinguishable, to OSCC portion (left side of the image)”. Hope the authors can explain further the features that are distinguishable between the healthy mucosa and the OSCC portion.
  2. Figure 4g: The "icicle-like" structures are missing which is indicative of OSCC. Hope the authors can explain this finding. 

Responses 8-9: Thank you for asking for further explanations on this figure, which we believe is truly representative. We have added further details in the text on the differences between the two portions of the Figure 4g and provided our interpretation of the absence of the ‘icicle-like’ structures; we believe that everything is more functional to a better understanding of the images.

  1. Figure 5 e,h,p: “fibrin deposits are indicated with a yellow rhombus”. Unfortunately, I am unable to appreciate the fibrin deposits in these photomicrographs. May be a photomicrograph of higher magnification of these regions can be provided as inserts within these figures. 

Response 10. Thank you for consideration. The presence of fibrin is a collateral finding of the neoplastic ulcer. We find it interesting that the ulceration areas are visible in the OCT images and that they can also be detected in the images of the histological sections. However, following your considerations, we have removed the indication of fibrin from the histological images (yellow rhombus in Figure 5 e,h,p), because it is not actually clearly evident at the photomicrographic magnification used.

Discussion

  1. “Already previous studies, applying OCT in oral oncology, have reported the loss of BM as a discriminatory criterion between healthy and cancerous tissues”. Kindly merge this sentence with the paragraph below. Kindly give reference by citing Obade et al 2021 and other articles for this sentence. 

Response 11. Thank you. Everything suggested has been done.

  1. “In addition, we believe that proposing a method of standardization of OCT-OSCCs expressed in specific descriptive characteristics (as standardized as possible), rather than in measurements of tissue layers (difficult to reproduce), can allow using the device more easily, even from non-oral medicine specialists”. Kindly change it to “for other specialist other than oral medicine”. 

Response 12: Thank you. The text has been changed as required

Additionally, a complete minor revision of the English, as requested, was performed.

Reviewer 3 Report

This pilot study is well planned, data are collected in a clear way and every paragraph is properly written. The paper is suitable for publication. 

Is an appropriate abstract included? 

Yes, it is.

                Is it clear what questions the paper is attempting to answer? 

                Are the objectives of the work clearly stated? 

Yes, it is.

                Are the methods clearly described? 

Yes, they are.

                Does the paper provide new knowledge, either in the way of evidence or interpretation to what is already known in the field? 

Yes, it does

                Does the paper discuss an issue of current concern in the field? 

Yes, it does

                Is the paper suitable for an international readership? 

Yes, it is

                Are the arguments sound? 

Yes, they are

                Are the experimental data capable of supporting the conclusions drawn? 

Yes, but it will probably be necessary to continue the study by including more cases

                Are there gaps or omissions in the coverage, data, logic or presentation? 

No, there are not.

                Is the paper well written and the data clearly presented by means of appropriate tables, graphs, or diagrams? 

Yes, the paper is well written. A future study, perhaps involving several research centers, could be useful to investigate this interesting topic in more depth and collect more data.

Author Response

Response to Reviewer 3 Comments

Review report:

This pilot study is well planned, data are collected in a clear way and every paragraph is properly written. The paper is suitable for publication.

  • Is an appropriate abstract included?

Yes, it is.

  • Is it clear what questions the paper is attempting to answer?
  • Are the objectives of the work clearly stated?

Yes, it is.

  • Are the methods clearly described?

Yes, they are.

  • Does the paper provide new knowledge, either in the way of evidence or interpretation to what is already known in the field?

Yes, it does

  • Does the paper discuss an issue of current concern in the field?

Yes, it does

  • Is the paper suitable for an international readership?

Yes, it is

  • Are the arguments sound?

Yes, they are

  • Are the experimental data capable of supporting the conclusions drawn?

Yes, but it will probably be necessary to continue the study by including more cases

  • Are there gaps or omissions in the coverage, data, logic or presentation?

No, there are not.

  • Is the paper well written and the data clearly presented by means of appropriate tables, graphs, or diagrams?

Yes, the paper is well written. A future study, perhaps involving several research centers, could be useful to investigate this interesting topic in more depth and collect more data.

 Response:

Thank you for your kindly detailed answers and for your useful suggestions. We are happy to inform you that the increase in the sample size, as well as the standardization of the methodological sequence, also for other pathologies of the oral cavity (in particular, potentially malignant disorders), is already active. Furthermore, we are forwarding the request for expression of interest in study participating to various national and international Research Units in Oral Medicine, hoping for broad membership.

A complete minor revision of the English, as requested, was performed.